# Lang2LTL: Translating Natural Language Commands to Temporal Specification with Large Language Models

Jason Xinyu Liu[1], Ziyi Yang[1], Benjamin Schornstein[1], Sam Liang[2], Ifrah Idrees[1], Stefanie Tellex[1] and Ankit Shah[1]

[1]Department of Computer Science, Brown University
[2]Department of Computer Science, Princeton University

**Abstract:** Robotic systems interacting with humans through natural language must adequately represent a wide range of challenging tasks. Linear temporal logic (LTL) has become a prevalent specification language to represent challenging non-Markovian tasks, such as completing subtasks in a specific order and repetitive task execution. In this work, we frame the problem of grounding natural language commands to an LTL expression as a neural machine translation problem, leveraging the capabilities of pre-trained large language models (LLMs). A key challenge for translation tasks is the collection of a large corpus of paired language and translated specifications. LLMs have demonstrated few-shot learning capabilities in many natural language tasks and can be used to overcome data-complexity challenges. We propose Lang2LTL, a new model architecture for translating natural language commands to LTL specifications. Results in navigation domains show that our modular approach outperforms our end-to-end baselines in translation accuracy and is more sample efficient than the encoder-decoder baseline at generalization across environments.

**Keywords:** Robots, Language, Linear Temporal Logic

## 1 Introduction

Commanding robotic systems through natural language represents a promising interaction modality for taskable multi-purpose robotic systems in the future. For example, human users can command an autonomous drone to always avoid certain air spaces while visiting a sequence of landmarks in a specific order and ask a robot to repetitively patrol regions of a school campus. Linear temporal logic (LTL) [1] is a widely used specification language that can express a range of challenging non-Markovian tasks, such as completing subtasks in a specific order, defining avoidance states, and repetitive task execution. This work proposes to ground natural language commands given to a robot to LTL expressions which serve as a task specification for downstream planning and learning algorithms. Following prior works [2, 3], we model this grounding procedure as a neural machine translation problem.

Existing methods [2, 3] using RNN encoder-decoder models for translation are only applicable within the environment the model was trained on and require retraining in every new environment. We take advantage of the recent success of large language models (LLMs) in many natural language tasks, such as name-entity recognition and machine translation. LLMs are generative models pretrained on a vast amount of data in varied linguistic contexts. We hypothesize that the large pre-trained corpus combined with a modular approach to deploying LLMs can outperform narrow environment-specific translation models developed in prior work in terms of sample efficiency and

6th Conference on Robot Learning (CoRL 2022), Auckland, New Zealand.

generalization ability. The key advantage of deploying pre-trained LLMs for command translation is its ability to transfer to a wide range of domains utilizing similar task descriptions, for example, navigation at multiple scales ranging from a single house to city-wide.

We examined three approaches for translating natural language commands to LTL specifications with varying levels of inductive bias and structure built into the model architecture. We propose decomposition of translation problem into subtasks, namely name-entity recognition (NER), grounding, and translation, where name entities in language commands are replaced by placeholder symbols during translation, and tested such a modular approach where each task is solved separately. We also propose two variants of the modular approach. In the first variant, the LLM is first called to solve the NER, and the grounding task. The model then translates the natural language command into LTL specification without explicitly replacing named entities in the commands with the place-holders. In the second variant the LLM is directly prompted to translate the input natural language command into an LTL formula without explicitly solving any of the subtasks. Results in navigation domains show that our modular approach outperforms our end-to-end baseline in translation accuracy and is more sample efficient across environments than a previously proposed encoder-decoder baseline.

## 2   Preliminaries

### 2.1   Linear Temporal Logic for Task Specification

Linear temporal logic (LTL) is a promising alternative to a numerical reward function for expressing task specifications. An LTL formula $\varphi$ is a Boolean function that determines whether a given trajectory has satisfied the objective expressed by the formula. Littman et al. [4] argue that such task specifications are more natural than numerical reward functions, and they have subsequently been used as a target language for acquiring task specifications in several settings, including from natural language [3] learning from demonstration [5] and generalization [6]. Formally, an LTL formula is interpreted over traces of Boolean propositions over discrete time, and is defined through the following recursive syntax:

$$\varphi := \alpha \mid \neg\varphi \mid \varphi_1 \vee \varphi_2 \mid \mathbf{X}\varphi \mid \varphi_1 \ \mathbf{U} \ \varphi_2 \tag{1}$$

Here $\alpha \in AP$ represents a Boolean proposition, mapping a state to a Boolean value; $\varphi$, $\varphi_1$, $\varphi_2$ are any valid LTL formulas. The operator $\mathbf{X}$ (next) is used to define a property $\mathbf{X}\varphi$ that holds if $\varphi$ holds at the next time step. The binary operator $\mathbf{U}$ (until) is used to specify ordering constraints. The formula $\varphi_1 \ \mathbf{U} \ \varphi_2$ holds if $\varphi_1$ holds until $\varphi_2$ first holds at a future time instant. The operators $\neg$ (not) and $\vee$ (or) are identical to propositional logic operators. We also utilize the following abbreviated operators: $\wedge$ (and), $\mathbf{F}$ (finally or eventually), and $\mathbf{G}$ (globally or always). $\mathbf{F}\varphi$ specifies that the formula $\varphi$ must hold at least once in the future, while $\mathbf{G}\varphi$ specifies that $\varphi$ must always hold in the future.

### 2.2   Prompting with Large Language Models

Large language models (LLMs) are large neural networks pretrained on huge text corpora in an unsupervised fashion. LLMs have successfully demonstrated few-shot adaptation on a wide gamut of tasks when framed as a text completion problem [7]. Deploying an LLM involves prompting the model with a priming text, followed by the model generating a completion. Generating a well performing prompt involves the largest amount of engineering effort. A successful prompt typically enables in-context learning by providing task descriptions and examples for the model to follow.

For instance, the prompt to the LLM for one translation task may start with the task description, "Your task is to translate English utterances into linear temporal logic (LTL) formulas" followed by an example and another English utterance to be translated,

Utterance: go to A but avoid going through B
LTL: $\mathbf{F}\ A \wedge \mathbf{G}\ !\ B$

Utterance: go to A then go to B
LTL:

## 3 Related Work

Prior work proposed using RNN encoder-decoder models to translate natural language commands to LTL expressions [2, 3, 8] that require training on a parallel corpus for every environment. Our approaches use LLMs to help generalize across environments. Patel et al. [3] introduced a weakly supervised approach where a latent LTL expression is fed into a planner that generates a trajectory to provide training signals. After examining the dataset and the model, we discovered that the intermediate LTL expressions mostly represent simple sequential tasks. We work on more diverse datasets that cover a wider range of LTL expressions. To improve the generalizability of encoder-decoder models to new environments, Berg et al. [8] used CopyNet [9] to resolve novel landmark names. However, the performance decreased significantly when there are more than one unseen landmarks. We used LLMs for name-entity recognition which showed better performance at identifying novel landmarks in input utterances.

Early works in navigational language grounding focused on utilizing semantic parsers to ground natural language into representations and then into robot actions [10, 11, 12, 13, 14]. Mei et al. [15] introduced a Bi-LSTM model that directly translated instructions into low-level actions. Fried et al. [16] applied pragmatic inference to further improve the performance. Neural end-to-end methods require the input utterance to be descriptive of low-level actions. In contrast, LTL expressions can model high-level language instructions describing complex robot behaviors and provide safety guarantees.

## 4 Translating Natural Language to LTL Specification

Consider a natural language command such as "The robot has to go to the cafe on Main street, then stop by a bank, then go to McDonald's, but only after visiting the bank", and a set of landmarks $\{Starbucks, Chase, McDonald, ...\}$ as propositions for constructing a valid LTL expression. The desired LTL formula of this command can be expressed as follows:

$$\varphi = \mathbf{F}\ (Starbucks \wedge \mathbf{F}\ Chase) \wedge \mathbf{F}\ McDonalds \wedge \neg McDonalds\ \mathbf{U}\ Chase \tag{2}$$

We decompose such translation of language to LTL specifications into the following subproblems:

1. Identification of substrings referring to Boolean propositions, in this case, "the cafe on Main street", "McDonald's" and "bank", but not "the robot".

2. Grounding referring substrings to environment propositions, i.e., knowing that the phrases "the cafe on Main street", "the bank" and "McDonald's" refer to the pre-defined environment propositions $Starbucks$, $Chase$, $McDonalds$.

3. Translating the input instruction into the LTL formula and simultaneously swapping the identified substrings into corresponding environment propositions.

In this section, we propose a modularized pipeline with pretrained large language models for tackling each of these subproblems and completing the translation of natural language to LTL specifications for the robotic system in the context of navigation, where the language propositions are grounded to known landmarks within the task environment.

We hypothesize that this task decomposition should produce a higher translation accuracy as compared to an approach that translates an input utterance directly into an LTL formula through a single prompt completion problem given to an LLM.

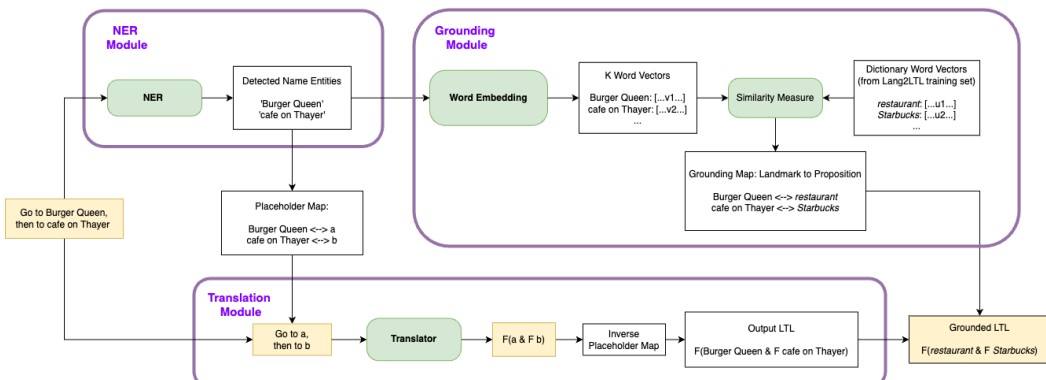

Figure 1: Lang2LTL-Modular Approach: Green blocks are pretrained or off-the-shelf models. Yellow blocks are input or output data of each module.

## 4.1   Name-Entity Recognition for Landmarks

In order to extract word sequences from given utterances, we add a name-entity recognition (NER) module based on a separate call to a pretrained LLM. Moreover, the task here is dissimilar to the typical NER task where all name entities are extracted and labeled. Instead, an NER module in our proposed pipeline must only output name entities with the required type as mentioned previously, and include informative words into each entity, e.g., "the cafe on Main street" rather than "the cafe" and "Main street".

The former can be solved by only selecting entities with *location* labels, while the latter is generally challenging to all existing pretrained NER models, especially without adequate examples for fine-tuning. We demonstrate high performance on the latter task by adapting the GPT-3 prompt with task description and examples to enable in-context learning. An example prompt is shown as follows,

> Your task is to repeat exact strings that refer to landmarks from the given utterance.
>
> Utterance: go to the bank then go to the restaurant
> Landmarks: the bank; the restaurant
>
> Utterance: go to the cafe on Main street, then a bank, then McDonald's
> Landmarks:

## 4.2   Grounding Name Entities to Environment Propositions

Due to the diversity of natural language, a single landmark can be referred to using multiple expressions. Grounding these expressions to the correct propositions is challenging. We approach this by labeling propositions with unique identifying language token. The problem of grounding expressions to propositions is then equivalent to mapping the expression to the most similar token.

We propose leveraging the latent embeddings computed by LLM encoders as the metric space for measuring similarity. Following Berg et al. [8], we match the referring expressions to the proposition tokens by matching their respective embeddings using the cosine similarity metric. This approach enables us to handle a variety of referring expressions, as the entire substring, e.g., attributes such

as "building" and "facility" or physical addresses such as "99 Main street", are transformed into the latent embedding by the LLM encoder.

### 4.3 Abstract Translation with Placeholders

A straight-forward way of translating natural language to LTL is to treat the language utterances as input and target LTLs as output. However, language generation is inherently difficult partly due to the large size of the target vocabulary, and generating language with strict syntactical restrictions heavy the burden. Even if LLMs are capable of performing in-context generation tasks, having a larger vocabulary can significantly deteriorate the translation performance.

In our modular approach, we add a translation module that replaces all the landmarks in a grounded language command with placeholder tokens (for example, letters of an alphabet). A separate call to the LLM with a prompt, including examples of commands and corresponding LTL formulas with the placeholders, is used to obtain the structure of the output LTL formula. An example of such prompt is shown in Section 2.2. Finally, the mapping from placeholder propositions to the substrings and the mapping from substrings to the environment propositions are used to transform the output LTL formula into the requisite output.

### 4.4 Proposed Lang2LTL model architectures

Our proposed modular approach, *Modular-NER+Placeholders*, is shown in Figure 1. The NER module takes natural language utterance as input and output identified landmark entities, and then the grounding module grounds each entity into a proposition, and then the translation module performs abstract translation with the placeholder map built on landmark entities and generates the grounded LTL expression as the final output.

We hypothesize that the key benefit of our modular approach is the decoupling of the problem of proposition recognition from the formula structure translation, thus allowing the language model to operate over a wide variety of LTL formula templates.

In addition to the modular approach, we also present two approaches in which the model makes a trade-off between inductive bias and generalization:

1. **Naive Translation** is a modular approach with a module that directly translates utterances with environmental propositions to final LTLs. This approach is proposed for evaluating the influence of abstract translation with placeholders, and it demands three calls to an LLM, the same as the modular approach. We refer to this approach as *Modular-NER* in subsequent sections.

2. **End-to-End Approach** takes as input a natural language command and a list of known landmarks situated in the environment and produces the corresponding LTL expression where the propositions are grounded to known landmarks. The approach translates the language commands into an LTL specification with a single call to an LLM. We refer to this approach as *End-to-End* in subsequent sections.

## 5 Experiments

Our experiments are aimed to evaluate the following hypotheses:

- **H1:** A modular approach performs better than an end-to-end approach at translating natural language to LTL specification.

- **H2:** A modular approach combined with LLMs is better than an RNN encoder-decoder model at generalizing across different domains that share similar task descriptions.

A number of pre-trained language models can be used for each module, e.g., BERT [17] and GPT-3 [18]. We chose GPT-3 because it is one of the largest language models that was pretrained on a

vast amount of data in varied linguistic contexts, and it has shown state-of-the-art performance in many natural language tasks.

## 5.1 Cleanup World Dataset

We evaluated the three Lang2LTL approaches on a subset of an annotated corpus of natural language commands paired with LTL expressions collected by Gopalan et al. [2]. The task environment, first introduced in [19], is partitioned into distinctly colored rooms, and the language commands instruct a robot to move through rooms in a specific order while possibly avoiding certain rooms. We chose this dataset as our test set because of the diverse natural language commands and LTL types compared to similar datasets from other prior work. Our test set contains 697 natural language commands corresponding to 14 unique LTL expressions in 4 different types. Some example pairs of language instructions and LTL expressions are shown in Table 2.

## 5.2 OpenStreetMap Dataset

To evaluate the generalization capability of our approaches that use LLMs, we randomly sampled 50 data points of similar task descriptions from the OpenStreetMap (OSM) dataset presented in [20]. With the same model architecture and prompts, we hypothesize that the LLM approach can achieve better translation accuracy on the OSM dataset than the baseline RNN encoder-decoder method. This test set consists of 48 unique LTL expressions in 4 different types, one of which is not presented in the prompt to the translation module. The language commands contain a diverse number of landmark names, which increases the difficulty of translation. Some example pairs of language instructions and LTL expressions are shown in Table 4.

## 5.3 Evaluation

To evaluate the performance of different approaches, we compare the output LTL expression with the ground truth one, and report an accuracy as the percentage of the test data that the two expressions are a match. We use Spot library [21] to test the logical equivalence of the two LTL expressions and the syntax of the output LTL expressions. Spot does the equivalence check by first converting the output and ground truth LTL formulas $f$ and $g$ and their negation into four automata $A_f$, $A_{\neg f}$, $A_g$ and $A_{\neg g}$, then evaluating if that the multiplicative conjunctions $A_f \otimes A_{\neg g}$ and $A_{\neg f} \otimes A_g$ are empty.

# 6 Results and Discussion

The accuracy of *Modular-NER+Placeholders* approach on the Cleanup World dataset is 84.10%, significantly higher that of *Modular-NER* (75.50%) and *End-to-End* (78.65%). The results show that building inductive bias into the system pipeline helps improve translation performance, and the choice of inductive biases are important, thus supporting **H1**. Some examples of successful and failed translations of the *Modular-NER+Placeholders* model on the Cleanup World dataset are shown in Table 2 and 3, respectively.

Table 1: Summary of Model Accuracies

| Model | Accuracy |
|---|---|
| *End-to-End* | 78.65% |
| *Modular-NER* | 75.50% |
| *Modular-NER+Placeholders* | **84.10%** |

Next, we evaluated the *Modular-NER+Placeholders* system using GPT-3 on the OSM dataset, and got a 72% translation accuracy. Some examples of successful and failed translations of the *Modular-NER+Placeholders* model on the OSM dataset are shown in Table 4 and 5, respectively. Given that the baseline RNN encoder-decoder model [2] was only trained on the Cleanup World dataset, its

Table 2: Examples Successful Translations in Cleanup World

| Input Natural Language Command | Output and Ground Truth LTL Expression |
|---|---|
| move to the red room | $\mathbf{F}\ red\_room$ |
| go through blue room to green room | $\wedge\ \mathbf{F}\ blue\_room\ \mathbf{F}\ green\_room$ |
| go to the blue room but avoid the red room | $\wedge\ \mathbf{F}\ blue\_room\ \mathbf{G}\ !\ red\_room$ |
| go through red or yellow to get to green | $\mathbf{F}\ \wedge\ \vee\ red\_room\ yellow\_room\ \mathbf{F}\ green\_room$ |

Table 3: Examples Failed Translations in Cleanup World

| Input Natural Language Command | Ground Truth LTL Expression
Output LTL Expression |
|---|---|
| move to the red room and stop after entering | $\mathbf{F}\ red\_room$
$\mathbf{F}\ \wedge\ red\_room\ \mathbf{X}$ |
| go to the blue room via the green room | $\mathbf{F}\ \wedge\ green\_room\ \mathbf{F}\ blue\_room$
$\mathbf{F}\ \wedge\ \vee\ green\_room\ \mathbf{F}\ blue\_room$ |
| go from the blue room to the red room
and then go from the red room to the green room | $\mathbf{F}\ \wedge\ red\_room\ \mathbf{F}\ green\_room$
$\mathbf{F}\ \wedge\ blue\_room\ \mathbf{F}\ red\_room\ \wedge\ \mathbf{F}\ green\_room$ |
| enter the blue room without crossing the red room | $\wedge\ \mathbf{F}\ blue\_room\ \mathbf{G}\ !\ red\_room$
$\mathbf{F}\ \wedge\ blue\_room\ \mathbf{G}\ !\ red\_room$ |

Table 4: Examples Successful Translations in OSM

| Input Natural Language Command | Output and Ground Truth LTL Expression |
|---|---|
| find science library | $\mathbf{F}\ science\_library$ |
| go to fedex office and then go to cvs | $\mathbf{F}\ \wedge\ fedex\_office\ \mathbf{F}\ cvs$ |
| stay away from main st and find chipotle | $\wedge\ \mathbf{G}\ !\ main\_st\ \mathbf{F}\ chipotle$ |
| stay on main st and find bookstore | $\wedge\ \mathbf{F}\ bookstore\ \mathbf{G}\ main\_st$ |

Table 5: Examples Failed Translations in OSM

| Input Natural Language Command | Ground Truth LTL Expression
Output LTL Expression |
|---|---|
| find bookstore and then find fedex office | $\mathbf{F}\ \wedge\ bookstore\ \mathbf{F}\ fedex\_office$
$\wedge\ \mathbf{F}\ bookstore\ \mathbf{GF}\ fedex\_office$ |
| go to citizens bank and then go to marston hall | $\mathbf{F}\ \wedge\ citizens\_bank\ \mathbf{F}\ marston\_hall$
$\mathbf{F}\ citizens\_bank\ \mathbf{F}\ marston\_hall$ |
| do not leave main st and find science library | $\wedge\ \mathbf{G}\ main\_st\ \mathbf{F}\ science\_library$
$\wedge\ \mathbf{G}\ !\ main\_st\ \mathbf{F}\ science\_library$ |

transfer ability to the OSM dataset is limited because the OSM contains more diverse language commands with a completely different set of named entities. This result supports **H2**.

We note that a large number of errors was due to the invalid syntax of the output LTL expressions, which shows the limited ability of GPT-3 to master LTL grammar with few-shot prompts. In addition, GPT-3 tends not to understand negation well, e.g., the translation of "go through the room which is not red to get to the blue room" is $\mathbf{F}\ \wedge\ red\_room\ \mathbf{F}\ blue\_room$ while the ground truth translation is $\wedge\ \mathbf{F}\ blue\_room\ \mathbf{G}!\ red\_room$.

# 7    Conclusion & Future Work

We introduced Lang2LTL, a novel model architecture that translates natural language commands with temporal specifications to LTL expressions, and showed that decomposing this task into sub-tasks and using an LLM for each module performs better than end-to-end approaches. We demonstrated the ability of LLM approaches to generalize across domains without retraining on parallel corpora. Lang2LTL enables human users to interact with robots via natural language commands containing complex temporal specifications.

For future work, there are several fruitful directions to explore: (1). Handling longer LTL specification generation tasks is beneficial to implementations in real scenarios, and could be potentially realized by further task decomposition with LLMs. (2). Resolving ambiguity in natural language, e.g., does "Go to the red room and then to the blue room" restrict the agent not to visit the blue room before reaching the red room? (3). In robotic applications, LTL formulas can be used to specify tasks for a planner, and the learned policy is executed on a real robot.

**Acknowledgments**

The authors would like to thank George Konidaris for his feedback to the project, reviewers for their comments, and the funding agencies for their support. This work is supported by NSF Graduate Research Fellowship Program funding and through a grant from Echo Labs.

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
