# OpenReview forum: "Lang2LTL: Translating Natural Language Commands to Temporal Specification with Large Language Models"
_robot-learning.org/CoRL/2022/Workshop/LangRob — LangRob 2022 Poster_

### Official Review · Reviewer_vq9R · 2022-11-09
**Major improvements needed**

**Rating:** 4
**Confidence:** 3

**Review:**

This paper presents an approach to decompose long natural language instructions into Linear temporal logic (LTL) expressions by prompting GPT-3. While interesting, I see two major issues with the manuscript: a) There are no robot experiments that show that the proposed approach helps robots in any downstream application b) It is also unclear why LTL should be the ideal interface for specifying tasks for robots. Recent works have shown that few-shot prompting of GPT-3 can help robots to directly write robot code [0,1], which seems like a more straightforward approach rather than having to translate LTL again into robot code. I would encourage having these approaches as baselines or discussing why LTL is better suited.

[0] Code as Policies: Language Model Programs for Embodied Control
[1] PROGPROMPT: Generating Situated Robot Task Plans using Large Language Models

---

### Official Review · Reviewer_cTqR · 2022-11-13
**Limited scope, qualitative and quantitative analysis.**

**Rating:** 4
**Confidence:** 4

**Review:**

The paper proposes a system that translates natural language commands with temporal specifications to LTL expressions. The results in navigation domains show that the proposed modular approach outperforms end-to-end baselines in translation accuracy and is more sample efficient than the encoder-decoder baseline at generalization across environments.

Weaknesses:
1. Motivation: Why is LTL a good representation of intermediate steps? The use of LTL allows for direct use of the task specification by the planner as well as for checkability and interpretability, however, recent papers have shown natural language to be powerful to represent intermediate steps by itself. It would be nice if the authors could present the necessary arguments for the use of LTL. The paper is missing important baselines where the intermediate steps are represented in natural language.
2. Limited analysis: Yes LLMs can be prompted to generate plans and LTL, but when do these models fail? What kinds of sequences do they have an easier time decoding versus which ones are harder?
3. Difficulty of tasks: From the examples presented in the paper, the complexity of the set of natural language instructions tested on and the resulting LTL expressions seems quite limited. Further lack of any explanation on what makes the datasets challenging and what are the reasons behind the differences in performances of the models seriously limits the contributions of this paper. Why does Modular-NER do worse than end-to-end?

---

### Decision · Program_Chairs · 2022-11-15

Accept (Poster)